

# Spatial heterogeneity effects on land surface modeling of water and energy partitioning

Lingcheng Li[1], Gautam Bisht[1], L. Ruby Leung[1]

[1] Atmospheric Sciences and Global Change Division, Pacific Northwest National Laboratory, Richland, WA, USA

Correspondence to: Lingcheng Li (lingcheng.li@pnnl.gov)





**Abstract**
Understanding the influence of land surface heterogeneity on surface water and energy fluxes is
crucial for modeling earth system variability and change. This study investigates the effects of four
dominant heterogeneity sources on land surface modeling, including atmospheric forcing (ATM),
soil properties (SOIL), land use and land cover (LULC), and topography (TOPO). Our analysis
focused on their impacts on the partitioning of precipitation (P) into evapotranspiration (ET) and
runoff (R), partitioning of net radiation into sensible heat and latent heat, and corresponding water
and energy fluxes. A set of 16 experiments were performed over the continental U.S. (CONUS)
using the E3SM land model (ELMv1) with different combinations of heterogeneous and
homogeneous datasets. The Sobol' total sensitivity analysis is utilized to quantify the relative
importance of the four heterogeneity sources. Results show that ATM and LULC are the most
dominant heterogeneity sources in determining spatial variability of water and energy partitioning,
and their heterogeneity effects are complementary both spatially and temporally. The overall
impacts of SOIL and TOPO are negligible, except TOPO dominates the spatial variability of R/P
across the transitional climate zone between the arid western and humid eastern CONUS.
Comparison with ERA5-Land reanalysis reveals that accounting for more heterogeneity sources
improves the simulated spatial variability of water and energy fluxes. An additional set of 13
experiments identified the most critical components within the heterogeneity sources: precipitation,
temperature and longwave radiation for ATM, soil texture and soil color for SOIL, and maximum
fractional saturated area parameter for TOPO.



## 1. Introduction

**Land surface heterogeneity plays a critical role in the terrestrial water, energy, and biogeochemical cycles from local to continental and global scales** (Giorgi and Avissar, 1997; Chaney et al., 2018; Zhou et al., 2019; Liu et al., 2017). As the land component of global Earth System Models (ESMs) and Regional Climate Models (RCMs), land surface models (LSMs) are used to simulate the exchange of momentum, heat, water, and carbon between land and atmosphere. LSMs have been widely utilized in studies focused on climate projection, weather forecast, flood and drought forecast, and water resources management (Clark et al., 2015; Lawrence et al., 2019). At the resolutions typically applied in ESMs and RCMs, LSMs have limited ability to resolve land surface heterogeneity to skillfully represent its impacts on the surface fluxes and subsequent effects on earth system and climate simulations through land-atmosphere interactions. Singh et al. (2015) demonstrated that increasingly capturing topography and soil texture heterogeneity at finer resolutions improves the land surface modeling of soil moisture, terrestrial water storage anomaly, sensible heat, and snow water equivalent. Therefore, better representing spatial heterogeneity in land surface modeling may be crucial to reliably simulate water and energy exchange between land and atmosphere (Essery et al., 2003; Jr. et al., 2017; Fan et al., 2019; Fisher and Koven, 2020).

**Several approaches have been developed to resolve land surface heterogeneity in LSMs.** The most common class of method is the tile approach that subdivides each grid into several tiles to account for heterogeneous surface properties (Avissar and Pielke, 1989). The Community Land Model version 5 (CLM5) and the Energy Exascale Earth System Model (E3SM) land model (ELM) utilize a nested subgrid hierarchy in which each grid cell is composed of multiple land units, soil





columns, and plant functional types. Tesfa et al. (2017; 2020) developed a topography-based
subgrid structure based on topographic properties such as surface elevation, slope, and aspect to
better represent topographic heterogeneity in ELM. Swenson et al. (2019) introduced hillslope
hydrology in CLM5 where each grid cell is decomposed into one or more multicolumn hillslopes.
The second class of method to account for land surface heterogeneity is called the "continuous
approach" in which subgrid heterogeneity is described via analytical or empirical probability
density functions (PDFs) instead of dividing a grid cell into subgrid units. For example, He et al.
(2021) developed the Fokker-Planck Equation subgrid snow model in the Rapid Update Cycle
Land-Surface Model, which uses dynamic PDFs to represent the variability of snow in each grid
cell. The third class of method to better account for land surface heterogeneity is by developing
parameterizations for subgrid processes. For example, Hao et al. (2021) implemented a sub-grid
topographic parameterization in the ELM to represent topographic effects on insolation, including
the shadow effects and multi-scattering between adjacent terrains. Besides these three classes of
approach dealing with subgrid heterogeneity, the fourth class is to directly increase the grid
resolution. Previous studies have demonstrated the benefits of increasing resolution in simulating
precipitation, temperature, and related extreme events over multiple spatial scales (Torma et al.,
2015; Lindstedt et al., 2015; Cuesta-Valero et al., 2020; Koster et al., 2002; Vegas-Cañas et al.,
2020; Rummukainen, 2016). The proposed hyperresolution land surface modeling by Wood et al.
(2011) to model land surface processes at a horizontal resolution of 1 km globally and 100 m or
finer continentally or regionally has been gaining attention as supported by increasing availability
of high performance computing resources  (Singh et al., 2015; Rouf et al., 2021; Ko et al., 2019;
Xue et al., 2021; Yuan et al., 2018; Chaney et al., 2016; Naz et al., 2018; Vergopolan et al., 2020;
Garnaud et al., 2016; Bierkens et al., 2014).






**There are several sources of heterogeneity in LSMs but their impact on water and energy**
**simulations at different spatial resolutions has not been systematically examined.** Four types
of heterogeneity sources are commonly categorized in land surface modeling, including
atmospheric forcing, soil properties, land use and land cover, and topography characteristics
(Singh et al., 2015; Ji et al., 2017). Singh et al. (2015) showed that including more detailed
heterogeneity of soil and topography at high resolutions improved the water and energy
simulations over the Southwestern U.S. Xue et al. (2021) demonstrated that simulations over the
High Mountain Asia region driven by high-resolution atmospheric forcing generally outperform
simulations that used coarse-resolution atmospheric forcing. Simon et al. (2020) investigated the
impacts of different heterogeneity sources (e.g., river routing and subsurface flow, soil type, land
cover, and forcing meteorology) on coupled simulations using the Weather Research and
Forecasting (WRF) model. They found that heterogeneous meteorology is the primary driver for
the simulations of energy fluxes, cloud production, and turbulent kinetic energy. Chaney et al.
(2016) conducted high-resolution simulations over a humid watershed and found that topography
and soils are the main drivers of spatial heterogeneity of soil moisture. However, these studies
generally focused either solely on one or few heterogeneity sources, or were conducted over small
domains with limited climate and hydrologic variations. Therefore, a comprehensive assessment
of the contribution of different heterogeneity sources to heterogeneity in energy and water fluxes
simulated by land surface models at continental scales is needed.

**The relative importance of heterogeneity sources on LSM simulations can be quantified by**
**sensitivity analysis (SA), which has been commonly used to study parametric uncertainty**





(Saltelli, 2002). In a quantitative sensitivity analysis, the assessed factors could include model
parameters as well as any other types of uncertainty induced by varying the input data (Saltelli et
al., 2019). The Sobol' global sensitivity analysis method is a variance-based SA approach and has
been widely utilized by the land surface modeling community (Rosolem et al., 2012; Nossent et
al., 2011; Li et al., 2013b). The most common application is assessment of model parameters
importance. Cuntz et al. (2016) comprehensively assessed the sensitivities of the Noah-MP land
surface model to selected parameters over 12 U.S. basins. This method is also utilized to quantify
the sensitivity of model outputs to the choice of parameterization schemes. Dai et al. (2017)
proposed a method based on Sobol' variance analysis to conduct sensitivity analysis while
simultaneously considering parameterizations and parameters. Zheng et al. (2019) utilized the
Sobol' method to quantify the sensitivity of evapotranspiration and runoff to different
parameterizations in the Noah-MP land surface model over the CONUS. Given the demonstrated
usefulness of the Sobol' sensitivity analysis method, it can be applied it to quantify the relative
importance of different heterogeneity sources on land surface water and energy simulations.

**The overarching goal of this paper is to determine the relative importance of different**
**heterogeneity sources on the spatial variability of simulated water and energy partitioning**
**over CONUS.** Four heterogeneity sources are considered in this study, including atmospheric
forcing (ATM), soil properties (SOIL), land use and land cover (LULC), and topography (TOPO).
Our analysis focuses on their impacts on the water partitioning of precipitation into
evapotranspiration and runoff, and the energy partitioning of net radiation into sensible heat and
latent heat, and their corresponding fluxes. ELMv1 is used as the model testbed. Two sets of
experiments are conducted with different combinations of homogeneous and heterogeneous inputs.





A set of 16 experiments are used to assess the impacts of the four heterogeneity sources on water
and energy partitioning using the Sobol' sensitivity analysis method. Subsequently, another set of
13 experiments are conducted to analyze the heterogeneity effects from each component of
atmospheric forcing, soil properties, and topography. The remaining structure of this paper is
organized as follows. Section 2 describes ELM, data processing, experimental design, and analysis
method. Results are examined in section 3, followed by discussions in section 4 and conclusions
in section 5.
**2. Methodology**
**2.1 ELM overview**
The E3SM is a newly developed state-of-the-science Earth system model by the U.S. Department
of Energy (Golaz et al., 2019; Caldwell et al., 2019; Leung et al., 2020). ELMv1 started from the
Community Land Model version 4.5 (CLM4.5; Oleson et al., 2013) and now includes more
recently developed representations of soil hydrology and biogeochemistry, riverine water, energy
and biogeochemistry, water management (2013a Li et al., 2013; Tesfa et al., 2014; Bisht et al.,
2018; Yang et al., 2019; Zhou et al., 2020). Further model developments after the ELMv1 release
include subgrid topographic parameterizations for solar radiation (Hao et al., 2021), a subgrid
topography structure (Tesfa and Leung, 2017) with subgrid downscaling of atmospheric forcing
(Tesfa et al., 2020), and plant hydraulics (Fang et al., 2021). However, these new developments
are not included in this study.

**2.2 ELM inputs**
**2.2.1 Heterogeneity sources**





ATM forcing for ELM consists of seven surface meteorological variables, including precipitation
(PRCP), air temperature (TEMP), specific humidity (HUMD), shortwave radiation (SRAD),
longwave radiation (LRAD), wind speed (WIND), and air pressure (PRES). Atmospheric forcing
from the North American Land Data Assimilation System phase 2 (NLDAS) is used in this study
(Xia et al., 2012b, a). SOIL consists of soil texture (STEX), organic matter content (SORG), and
soil color (SCOL). STEX and SORG determine soil thermal and hydrologic properties, while
SCOL regulates the soil albedo and hence surface energy related processes. LULC consists of the
glacier, lake, and urban fractions, the fractional cover of each plant functional type (PFT), and
monthly leaf area index (LAI) and stem area index (SAI) for each PFT. The high-resolution
datasets of land use land cover, leaf area index, and stem area index at 0.05°×0.05° developed by
Ke et al. (2012) are used for LULC in this study. TOPO consists of the standard deviation of
elevation (SD_ELV), maximum fractional saturated area (Fmax), and topography slope. TOPO is
used in snow cover parameterization, surface runoff generation and infiltration, etc. SOIL and
TOPO datasets are obtained from the NCAR dataset pool for CLM5 (Lawrence et al., 2019;
Lawrence and Chase, 2007; Bonan et al., 2002; Batjes, 2009; Hugelius et al., 2013; Lawrence and
Slater, 2008). Table 1 summarizes these heterogeneity components and resolutions of the source
data. All datasets were prepared over the entire CONUS.

161            Table 1 Summary of heterogeneity sources in ELM model inputs

| Heterogeneity source | Components | Source data resolution |
|---|---|---|
| ATM | Precipitation, air temperature, specific humidity, shortwave radiation, longwave radiation, wind speed, air pressure | 0.125°, hourly |
| SOIL | Soil texture, soil organic matter | 0.083°, static |
|  | Soil color | 0.5°, static |
| TOPO | Slope, Standard deviation of elevation, maximum fractional saturated area | 0.125°, static |
|  | Fractions of PFTs, wetland, lake, urban characteristics, and glacier | 0.05°, static |
| LULC | Leaf area index (LAI) for each PFT | 0.05°, monthly |






### 2.2.2 Heterogeneous and homogeneous inputs

We prepared heterogeneous and homogeneous inputs at 0.125°×0.125°. The difference between the two datasets is whether the input values within each 1°×1° region of ELM are spatially heterogeneous or homogeneous. The four types of datasets listed in Table 1 were first resampled to 0.125°×0.125° resolution from their original resolutions, which are used as the heterogeneous inputs (Figures 1a and 1b). Then, for each dataset, we replaced the heterogeneous values of the 64 0.125°×0.125° grids within each 1°×1° region by the mean of the 64 grids (see Figure 1b vs. 1d). The temporally varying datasets (e.g., hourly ATM and monthly climatology LAI) were processed at each time interval. As an example, Figure 1 compares the annual climatology of the heterogeneous and homogeneous precipitation.

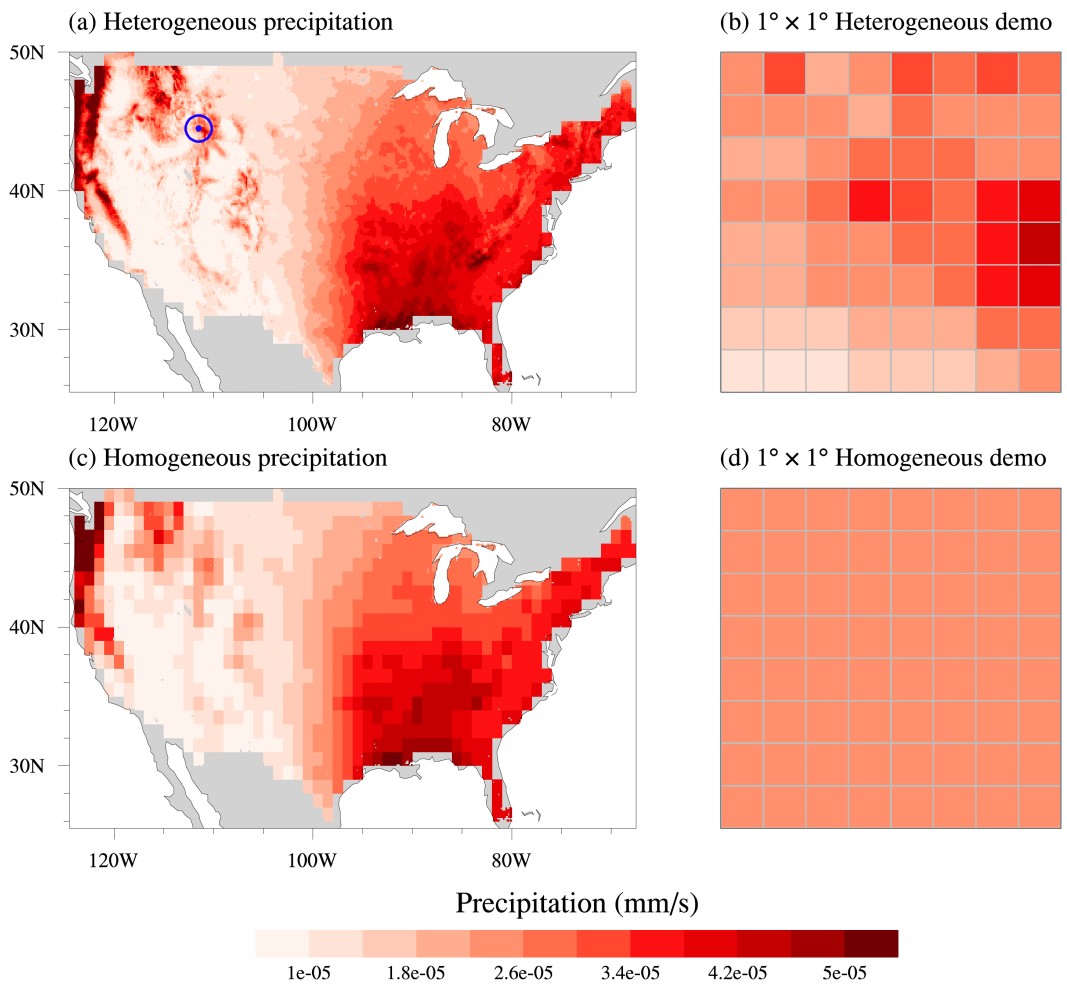


Figure 1. Annual climatology of (a) heterogeneous and (c) homogeneous precipitation over

CONUS. The corresponding (b) heterogeneous and (d) homogeneous precipitation over a 1°×1°

region (latitude: 37° N ~ 38° N, longitude: 111° W ~ 110° W, the blue marker in (a)) is also shown.


**2.3 Experimental design and analysis**

We conducted two sets of ELM experiments over CONUS. The first set contains 16 experiments

with different combinations of heterogeneous and homogeneous inputs from the four heterogeneity





sources (Table 2). These experiments were used to quantify the influence of different heterogeneity
sources on the ELM simulations. The second set of 13 experiments were further conducted to
analyze the impact of heterogeneity from individual components of three heterogeneity sources
(Table 3). As LULC has no explicit individual component, we only analyzed the components of
ATM with seven experiments, SOIL with three experiments, and TOPO with three experiments.
Each experiment only contains one heterogeneous input while other components are homogeneous.
Both the first and second set of experiments were configured at 0.125°×0.125° spatial resolution.
The 40-year NLDAS-2 forcing from 1980–2019 was cycled twice to drive the ELM run for 80
years. The first 50-year run was used as model spin-up, and the last 30-year simulation
(corresponding to atmospheric forcing from 1990–2019) was used for further analysis.

Table 2. The first set of 16 experiments with inputs from ATM, SOIL, LULC, and TOPO.
(0 and 1 denote homogeneous and heterogeneous input from the four heterogeneity sources,
respectively)

| No. | Abbr. | ATM | SOIL | LULC | TOPO |
|------|----------|-----|------|------|------|
| EXP1 | A0S0L0T0 | 0 | 0 | 0 | 0 |
| EXP2 | A0S0L0T1 | 0 | 0 | 0 | 1 |
| EXP3 | A0S0L1T0 | 0 | 0 | 1 | 0 |
| EXP4 | A0S0L1T1 | 0 | 0 | 1 | 1 |
| EXP5 | A0S1L0T0 | 0 | 1 | 0 | 0 |
| EXP6 | A0S1L0T1 | 0 | 1 | 0 | 1 |
| EXP7 | A0S1L1T0 | 0 | 1 | 1 | 0 |
| EXP8 | A0S1L1T1 | 0 | 1 | 1 | 1 |
| EXP9 | A1S0L0T0 | 1 | 0 | 0 | 0 |
| EXP10 | A1S0L0T1 | 1 | 0 | 0 | 1 |
| EXP11 | A1S0L1T0 | 1 | 0 | 1 | 0 |
| EXP12 | A1S0L1T1 | 1 | 0 | 1 | 1 |
| EXP13 | A1S1L0T0 | 1 | 1 | 0 | 0 |
| EXP14 | A1S1L0T1 | 1 | 1 | 0 | 1 |
| EXP15 | A1S1L1T0 | 1 | 1 | 1 | 0 |
| EXP16 | A1S1L1T1 | 1 | 1 | 1 | 1 |




Table 3. The second set of 13 experiments with inputs from each component of the heterogeneity

sources.

| No. | Sole heterogeneity input |
|---|---|
| **ATM** | |
| ATM1 | Precipitation |
| ATM2 | Air temperature |
| ATM3 | Specific humidity |
| ATM4 | Shortwave radiation |
| ATM5 | Longwave radiation |
| ATM6 | Wind speed |
| ATM7 | Air pressure |
| **SOIL** | |
| SOIL1 | Soil texture of sand, silt, and clay |
| SOIL2 | Soil organic matter |
| SOIL3 | Soil color |
| **TOPO** | |
| TOPO1 | Fmax |
| TOPO2 | Standard deviation of elevation |
| TOPO3 | Slope |


Our analysis focused on water partitioning, energy partitioning, and related flux variables. The
water partitioning is quantified as the ratio between evapotranspiration (ET) and precipitation (P),
i.e., ET/P, and the ratio between runoff (R) and precipitation (P), i.e., R/P. The energy partitioning
is quantified using the evaporative fraction (EF), which equals the ratio between latent heat (LH)
and the sum of latent heat and sensible heat (SH), i.e., $EF = LH/(LH + SH)$. Based on outputs
from each experiment, the 30-year monthly, seasonal, and annual climatological means were first
calculated at 0.125°×0.125° resolution for the five variables of interest (i.e., P, ET, R, LH, and SH).
Second, the water and energy partitioning variables (i.e., ET/P, R/P, EF) were computed at
0.125°×0.125° resolution. Third, the standard deviations (SD) of these variables were calculated
for each 1° × 1° region from the encompassed 64 0.125°×0.125° grids. These 1° x 1° resolution
SDs of the first and second set of experiments were used in following analysis.





For the first set of 16 experiments, we utilized the Sobol' sensitivity analysis to quantify the relative
importance of the four heterogeneity sources on water and energy simulations. Detail of the Sobol'
sensitivity analysis is described in section 2.4.
The Sobol' method was not used for the second set of 13 experiments because a comprehensive
Sobol' analysis needs $2^{13}$ experiments, which is computationally infeasible. Instead, the calculated
SD of each experiment is used to quantify the impact of heterogeneity of each component, as each
experiment only contains one heterogeneous input. Therefore, we compared the SDs between each
experiment to determine the relative importance of each component with heterogeneous input
(without considering interactions between different components).

**2.4 The Sobol' total sensitivity index**
The Sobol' sensitivity analysis (Sobol', 1993) was applied to quantify the sensitivity of spatial
variation (i.e., SD) of water and energy partitioning to the four heterogeneity sources based on the
first set of 16 experiments. The Sobol' total sensitivity index, $SI_{X_i}$, is given as,

$$SI_{X_i} = \frac{E_{X_{\sim i}}(V_{X_i}(Y|X_{\sim i}))}{V(Y)} \tag{1}$$

where $X_i$ ($i = 1,2,3,4$) is the $i$-th heterogeneity source (e.g., ATM, SOIL, LULC, and TOPO); $X_{\sim i}$
denotes the other heterogeneity factors except $X_i$; $Y$ represents the corresponding SDs for a given
simulated variable of all 16 experiments. $V(Y)$ is the total variance of all the 16 SDs. The SDs of
the 16 experiments are then reformed into 8 subgroups based on experiments with different
combinations of $X_{\sim i}$; $V_{X_i}(Y|X_{\sim i})$ denotes the variance of SDs of each subgroup of experiments
with heterogeneous and homogeneous inputs of $X_i$; $E_{X_{\sim i}}$ is the arithmetic average across different
combinations of heterogeneity sources other than $X_i$.





Table 4 demonstrates the calculation of the Sobol' index to quantify the sensitivity of EF spatial
variability to LULC in a 1° × 1° region at 39.5N and 107.5W. The 16 experiments are grouped
into eight subgroups containing two experiments, where the difference between the two
experiments in a given subgroup is homogeneous vs. heterogeneous LULC. The SDs of the 16-
experiments are listed in C1. The variance of each subgroup is computed in C2, which represents
the influence of LULC heterogeneity. The average impact of LULC heterogeneity from the eight
subgroups in C3 is computed as the mean of values in C2. The total variance of these 16 SDs in
C1 is computed in C4. Finally, the ratio between C3 and C4 is calculated as Sobol' total sensitivity
index in C5, which quantifies EF spatial variability sensitivity to LULC heterogeneity. The Sobol'
total sensitivity index for ATM, TOPO, and SOIL index can be computed similarly.

Table 4 Demonstration of Sobol' index calculation of the sensitivity of EF spatial
variability to LULC

| Experiments | $Y$ | $V_{LULC}(Y\|X_{\sim LULC})$ | $E_{\sim LULC}(V_{LULC}(Y\|X_{\sim LULC}))$ | $V(Y)$ | $SI_{LULC}$ |
|---|---|---|---|---|---|
| C0 | C1 | C2 | C3 | C4 | C5 |
| A0S0L0T0 | 0.00 | 6.88 | | | |
| A0S0L1T0 | 5.24 | | | | |
| A0S0L0T1 | 0.57 | 6.28 | | | |
| A0S0L1T1 | 5.58 | | | | |
| A0S1L0T0 | 0.32 | 6.75 | | | |
| A0S1L1T0 | 5.51 | | | | |
| A0S1L0T1 | 0.69 | 6.64 | | | |
| A0S1L1T1 | 5.84 | | 3.32 | 26.99 | 0.12 |
| A1S0L0T0 | 12.88 | 0.01 | | | |
| A1S0L1T0 | 12.67 | | | | |
| A1S0L0T1 | 12.80 | 0.00 | | | |
| A1S0L1T1 | 12.76 | | | | |
| A1S1L0T0 | 12.71 | 0.01 | | | |
| A1S1L1T0 | 12.51 | | | | |
| A1S1L0T1 | 12.63 | 0.00 | | | |
| A1S1L1T1 | 12.59 | | | | |


**2.5 ERA5-Land reanalysis dataset**



We further compared the first set of experiments with ERA5-land reanalysis (the land component
of the fifth generation of European Centre of Medium-range Weather Forecast reanalysis) (Muñoz-
Sabater et al., 2021) to demonstrate the added value in ELM simulations with consideration of
heterogeneity sources. ERA5-Land provides a consistent view of terrestrial water and energy
cycles at high spatial and temporal resolutions. The monthly ERA5-Land data at 0.1°×0.1°
resolution was used in this study. First, the monthly data was resampled to 0.125°×0.125°
resolution. Second, the annual and seasonal climatological means for related variables (e.g., ET,
R, SH) were computed. Third, SD for each variable was calculated within each 1°×1° region for
further comparisons with the ELM simulations.





**3. Results**
**3.1. CONUS overall heterogeneity sensitivities**
The inclusion of more heterogeneity sources leads to larger spatial variability in the simulated
ET/P, R/P, and EF (Figure 2). For example, comparing experiment A0S0L0T0 with A1S0L0T0
that includes the ATM heterogeneity, the CONUS averaged SD for ET/P increases from 0 to 4.7%
(Figure 2a). By further comparing experiments in the first and third rows with the second and
fourth rows, ATM always increases the spatial variability of water and energy partitioning.
Similarly, LULC heterogeneity also shows large impacts on the spatial variability for the
partitioning variables as indicated by comparing experiments in the first and third columns with
the second and fourth columns. However, heterogeneity in SOIL and TOPO show negligible
impact. The effects of the heterogeneity sources on the spatial variability of water and energy
partitioning are mainly located in western and central CONUS (Figure S1), which is consistent
with the spatial variability of the heterogeneity inputs, for variables such as precipitation, air
temperature, and longwave radiation (Figure S2).

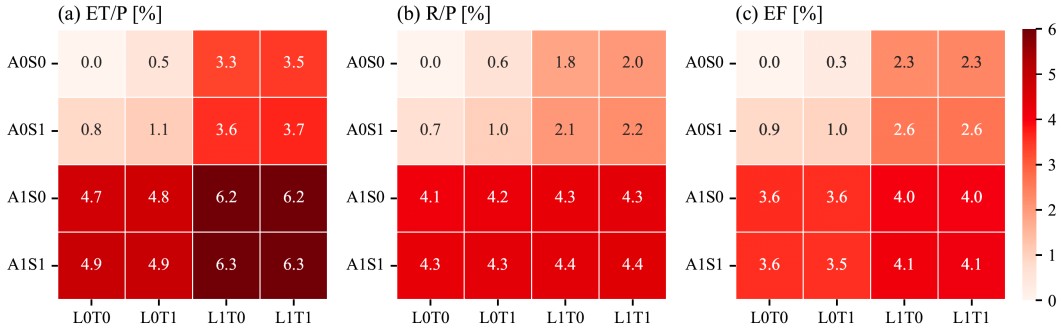


Figure 2. CONUS averaged SD of the annual climatology of (a) ET/P, (b) R/P, and (c) EF.
Combining the X-axis label for LULC and TOPO and the Y-axis label for ATM and SOIL
indicates the names of the experiments listed in Table 2, highlighting the use of heterogeneous
(1) and homogeneous (0) inputs for each heterogeneity source.






ATM, with the largest Sobol' sensitivity index, is the most important heterogeneity source to
determine the spatial variability of water and energy partitioning (Figure 3). LULC is the second
most important heterogeneity source. However, the heterogeneity of SOIL and TOPO marginally
contribute to the spatial variability of water and energy partitioning. Even though ATM dominates
the spatial heterogeneity of total ET, LULC is the main contributor to the spatial variability of the
ET components of transpiration, canopy evaporation, and ground evaporation. TOPO shows larger
impacts on the spatial variabilities of the runoff components than the total runoff.

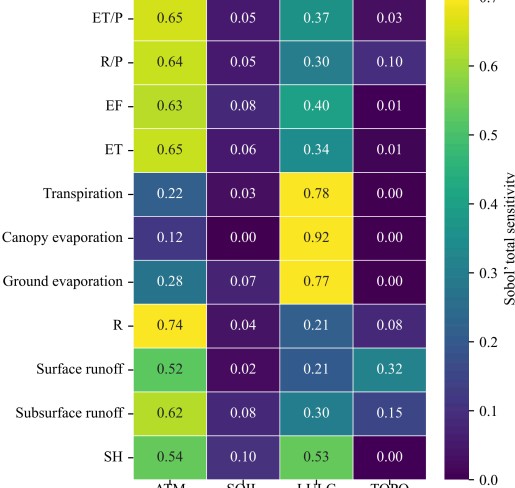


Figure 3. CONUS averaged Sobol' sensitivity index for the sensitivity of spatial variability of

different variables (rows) to the four heterogeneity sources (columns).

**3.2 Spatial patterns of heterogeneity sensitivities**
The sensitivity of the four heterogeneity sources shows different spatial patterns over CONUS
(Figure 4). The water partitioning components, ET/P and R/P, exhibit similar spatial patterns of
Sobol' sensitivity index for any given heterogeneity source (Figures 4a-d, 4f-i). ATM shows high



Sobol' sensitivity index over most CONUS regions for water and energy partitioning. It dominates
the spatial variability of ET/P and R/P over eastern and western CONUS but not central CONUS
(Figures 4e and 4j). For the spatial variability of EF, ATM mostly shows dominant effects over
central and western CONUS (Figures 4o). LULC is the second most dominant heterogeneity
source and dominates most regions over eastern CONUS (Figure 4o), although LULC also
dominates smaller regions for the spatial variability of ET/P and R/P over central and southeastern
CONUS (Figures 4e and 4j). Although TOPO overall has low Sobol' index, it dominates the spatial
variability of R/P over central CONUS (Figure 4j). SOIL has negligible impacts over most regions
of CONUS for the spatial variability of both water and energy partitioning.

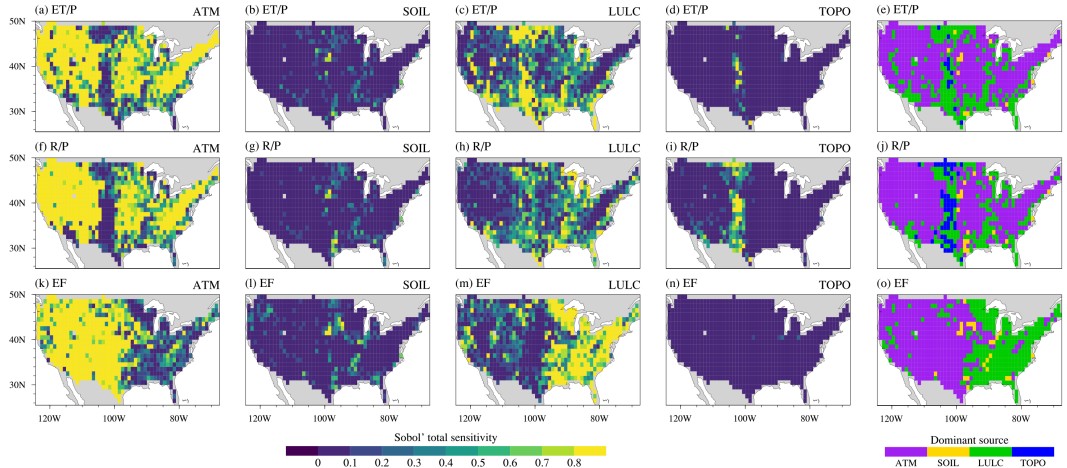


Figure 4. Spatial patterns of Sobol' total sensitivity index to the four heterogeneity sources
(column 1-4) and the corresponding dominant sources (column 5) for the spatial variability of
water (ET/P and R/P) and energy (EF) partitioning.

To further explain the spatial patterns of the Sobol' index for the two most dominant heterogeneity
sources of ATM and LULC, we further analyzed EXP9 (A1S0L0T0) and EXP3 (A0S0L1T0) listed



in Table 2. EXP9 and EXP3 only include heterogeneous inputs from ATM and LULC, respectively.
Let us consider ET/P as the quantity of interest for the following discussion. First, the SD of ET/P
is computed from the annual climatology (see section 2.3). Next, the SD ratio of ET/P, denoted as
$SDR_{ET/P}$, is computed as the ratio between the SD of ET/P in EXP9 and EXP3. $SDR_{ET/P}$
represents the relative spatial variability induced by ATM compared to LULC (Figure S3a). The
spatial pattern of the ATM Sobol' index for the ET/P spatial variability shows a positive
relationship with the spatial pattern of $SDR_{ET/P}$ (purple circles in Figure 5a, corresponding to
Figure 4a vs. Figure S3a). Therefore, a higher value of $SDR_{ET/P}$ indicates that relative to LULC,
ATM induces larger ET/P spatial variability, and hence has a higher ATM Sobol' index. Similarly,
a lower value of $SDR_{ET/P}$ indicates LULC induces larger ET/P spatial variability than ATM, and
hence has a higher LULC Sobol' index (green circles in Figure 5a). Similarly, $SDR_{R/P}$ and $SDR_{EF}$
were calculated for R/P and EF, and they also show a positive (negative) relationship with the
corresponding ATM (LULC) Sobol' index (Figures 5b and 5c, and Figures S3b and S3c). We can
also see that the ATM Sobol' index has opposite spatial patterns compared to that of the LULC
Sobol' index. Therefore, ATM and LULC show complementary contributions to the spatial
variability of water and energy partitioning across CONUS.

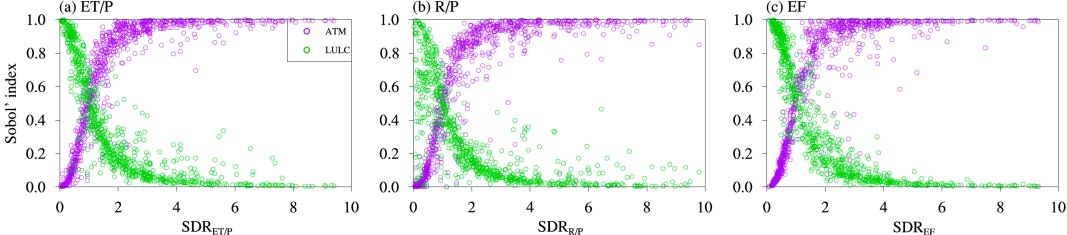

Figure 5. CONUS spatial relationship between the ATM and LULC Sobol' sensitivity index and
the SD ratio for (a) ET/P, (b) R/P, (c) EF. The y-axis values correspond to the spatial patterns of
the Sobol' index for ATM (purple) and LULC (green) in Figure 4 (i.e., each circle corresponds to





each 1°×1° region in Figure 4). The x-axis corresponds to the spatial pattern of the SD ratio in
Figure S3.

**3.3 Seasonal variation of heterogeneity sensitivities**
The impacts of ATM and LULC on the spatial variability of water and energy fluxes show more
seasonal variations than the impacts of SOIL and TOPO (Figure 6, SOIL and TOPO are not shown
here). This is because ATM and LULC consist of time-varying inputs to the ELM simulations, but
SOIL and TOPO are time-invariant inputs. Even though the spatial distribution of LULC is
temporally static, the monthly variations in LAI and SAI of different land cover types could affect
the seasonal variation of sensitivity. The heterogeneity impacts of ATM and LULC on the spatial
variability of water and energy fluxes show complementary seasonal variations. The effect of
ATM on the ET spatial variability is larger in July–September than in other months (Figure 6a),
while LULC shows smaller Sobol' index in July–September. The sensitivity of transpiration and
canopy evaporation shows the same seasonal variations (Figures S4a~c). The spatial variability of
R is more sensitive to ATM in the cold season (December–April, Figure 6b), especially for its
component of surface runoff (Figure S4d). The sensitivity of SH spatial variability to ATM is
larger in the non-growing season (i.e., November–March) than in the growing season (i.e., April–
October), with the LULC Sobol' index showing opposite seasonal variations.

To further explain the seasonal variations of the Sobol' index for ATM and LULC, the SD of ET
for each month was calculated as an example from monthly mean climatology and the SD ratio for
each month was computed as the ratio between the SD of ET in EXP9 and EXP3. For each 1°×1°
region, the 12 monthly SD ratios were normalized to [0, 1] using minimum and maximum values.



Finally, the CONUS average of the normalized SD ratios was computed for each month, denoted
as $NSDR_{ET}$. A higher value of $NSDR_{ET}$ denotes ATM induces more ET spatial variability than
LULC. Therefore, $NSDR_{ET}$ shows similar seasonal variations with the ATM Sobol' index for ET
spatial variability (black curve vs. purple curve in Figure 6a), but opposite seasonal variations with
the LULC Sobol' index (black curve vs. green curve in Figure 6a). Similarly, $NSDR_R$ and $NSDR_{SH}$
were calculated for R and SH, and they also show a similar (opposite) seasonal variation with the
corresponding seasonal ATM (LULC) Sobol' index (Figures 6b and 6c).

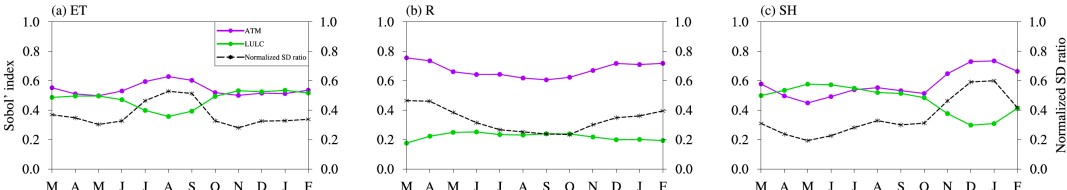

Figure 6. Monthly variations of CONUS averaged ATM and LULC Sobol' index and normalized
SD ratio for (a) ET, (b) R, and (c) SH.

The spatial patterns of dominant regions by the four heterogeneity sources vary over different
seasons. Compared with spring and winter, ATM dominates the ET spatial variability in more
regions than in summer and fall when ATM is more dominant over eastern CONUS (Table 5 and
Figures S5a~d). LULC shows opposite seasonal spatial patterns with more dominant regions in
eastern CONUS over spring and winter. As for the R spatial variability, TOPO shows large spatial
variation of its dominant regions over different seasons (Figures S5f~i). Besides its dominant
contribution in central CONUS over all seasons, TOPO also dominates the R spatial variability in
parts of eastern US in the summer and autumn (Figures S5g~h). For the EF spatial variability,
ATM has more contributions in the fall and winter but smaller contributions in spring and summer



than LULC (Table 5). LULC shows more dominant regions over eastern CONUS, especially in
spring and summer (Figures S5k~i).
Table 5 Grid percentage of the dominant heterogeneity source in determining the spatial
variability of ET, R, and SH for four seasons and annual mean (ANN)

| Seasons | ATM | SOIL | LULC | TOPO |
|---|---|---|---|---|
| ET | | | | |
| Spring (MAM) | 51 | 4 | 46 | 0 |
| Summer (JJA) | 63 | 3 | 34 | 0 |
| Fall (SON) | 57 | 2 | 42 | 0 |
| Winter (DJF) | 49 | 0 | 51 | 0 |
| ANN | 66 | 2 | 31 | 0 |
| R | | | | |
| Spring (MAM) | 81 | 2 | 13 | 5 |
| Summer (JJA) | 67 | 4 | 17 | 11 |
| Fall (SON) | 66 | 6 | 18 | 11 |
| Winter (DJF) | 75 | 2 | 12 | 10 |
| ANN | 77 | 1 | 15 | 7 |
| SH | | | | |
| Spring (MAM) | 44 | 5 | 51 | 0 |
| Summer (JJA) | 45 | 2 | 53 | 0 |
| Fall (SON) | 52 | 5 | 44 | 0 |
| Winter (DJF) | 69 | 2 | 29 | 0 |
| ANN | 49 | 4 | 47 | 0 |


**3.4 Effects of ATM heterogeneity components**

Based on the second set of 13 experiments, we analyzed the heterogeneity effects by each
component of ATM, SOIL, and TOPO (Figure 7), respectively. Precipitation is the largest ATM
heterogeneity source in determining the spatial variability of water fluxes (Figures S6a~b),
especially over western and central CONUS for ET (Figure 7a) and almost the entire CONUS for
R (Figure 7b). Air temperature dominates the spatial variability of ET in eastern CONUS (Figure
7a). The spatial variability of SH is mainly dominated by the incoming longwave radiation in
western CONUS and by the air temperature in eastern CONUS (Figure 7c). Longwave radiation
provides more energy input and contributes more to the SH spatial variability than shortwave
radiation (Figure S6c). Among the SOIL components, soil texture, which can influence soil





moisture and runoff generation, shows the largest effects on the ET and R spatial variability over
most CONUS regions (Figures 7d and 7e). Soil color, affecting the surface albedo and energy
balance, shows the largest impacts on the SH spatial variability over central CONUS (Figures 7f
and 8f). Fmax is the most essential TOPO component, offering the largest effects on the spatial
variability of ET, R, and SH over most CONUS regions (Figures 7g~i and Figures S6g~i). Fmax
regulates surface runoff generation and infiltration, and therefore influences the soil moisture, ET,
and SH. SD_ELV and slope can affect surface water and snow cover fraction, and consequently,
they show the largest impacts over northwestern CONUS regions with mountains and snowpack.
The spatial variability induced by all components (of ATM, SOIL, or TOPO) is larger than that
induced by each individual component. However, it is smaller than the sum of the spatial
variability induced by each component (Figure S6). For example, the CONUS averaged SD for
ET caused by all SOIL components is 1.9 ($10^{-7}$ mm/s), which is smaller than 2.5 ($10^{-7}$ mm/s), the
sum of the SD of ET induced by STEX, SORG, and SCOL (Figure S6d). Therefore, the additional
SD induced by an additional heterogeneity component decreases, suggesting that the effect of
heterogeneity on the spatial variability of water and energy fluxes saturates, possibly due to
interactions among the processes influenced by the heterogeneity sources.

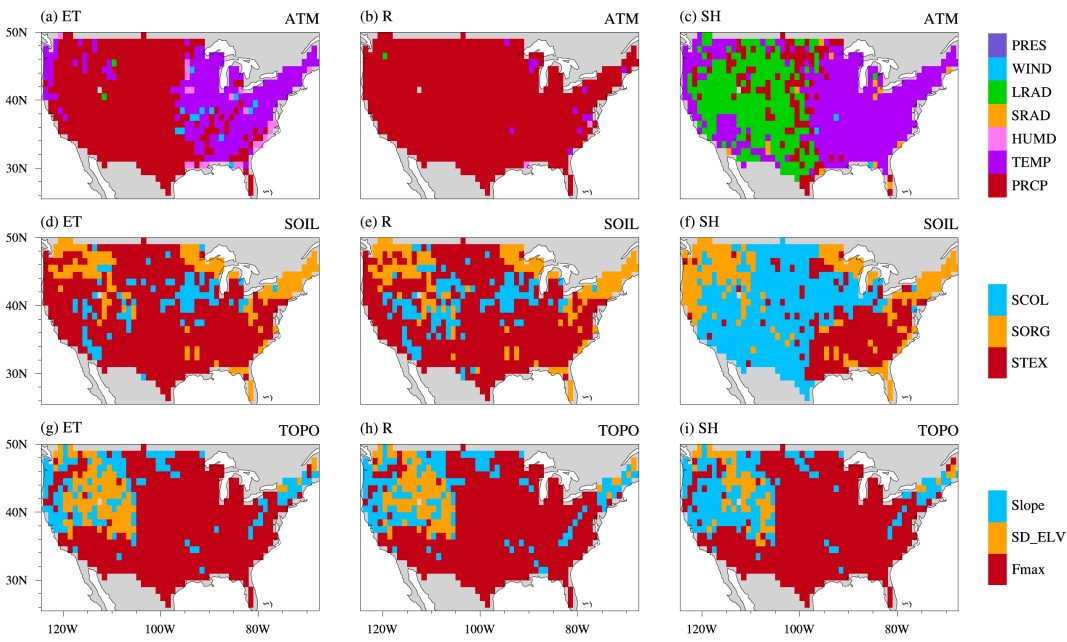


Figure 7. The largest induced spatial variability for the annual climatological mean of ET (left column), R (middle column), and SH (right column) induced by each component of ATM (top panel), SOIL (middle panel), and TOPO (bottom panel)


## 3.5 Comparison with ERA5-Land reanalysis

Higher consistency of the spatial variability between the simulations and ERA5-Land reanalysis (i.e., smaller SD difference) is obtained when more sources of heterogeneity are accounted for in the simulations for ET, R, and SH (Figure 8). ATM and LULC dominate the improvements of the spatial variability of model simulations. Generally, ATM heterogeneity leads to more or similar improvements than LULC heterogeneity for ET, R, and SH over all seasons. For example, in Figure 8a, ATM induced larger improvements, as shown by comparing experiments in the first and third rows with the second and fourth rows, than the LULC induced improvements, comparing experiments in the first and third columns with the second and fourth columns. The SD difference is usually larger over MAM and JJA than SON and DJF, probably due to the heterogeneity



difference between the NLDAS and ERA5 atmosphere forcing as ATM is the major heterogeneity
contributor. Improvements of the spatial variability of model simulations are primarily distributed
over western and eastern CONUS for ET and R, and western CONUS for SH (e.g., Figures S7 1st
column vs. 4th column). For ET and R, ATM mainly improves their spatial variability over western
and eastern CONUS (Figures S7a vs. S7c, and S7e vs. S7g), and LULC mostly shows
improvements over eastern CONUS (Figures S7a vs. S7b, and S7e vs. S7f). Both ATM and LULC
show improvements in the SH spatial variability over western and eastern CONUS (Figure S7i vs.
S7j, and S7i vs. S7k).

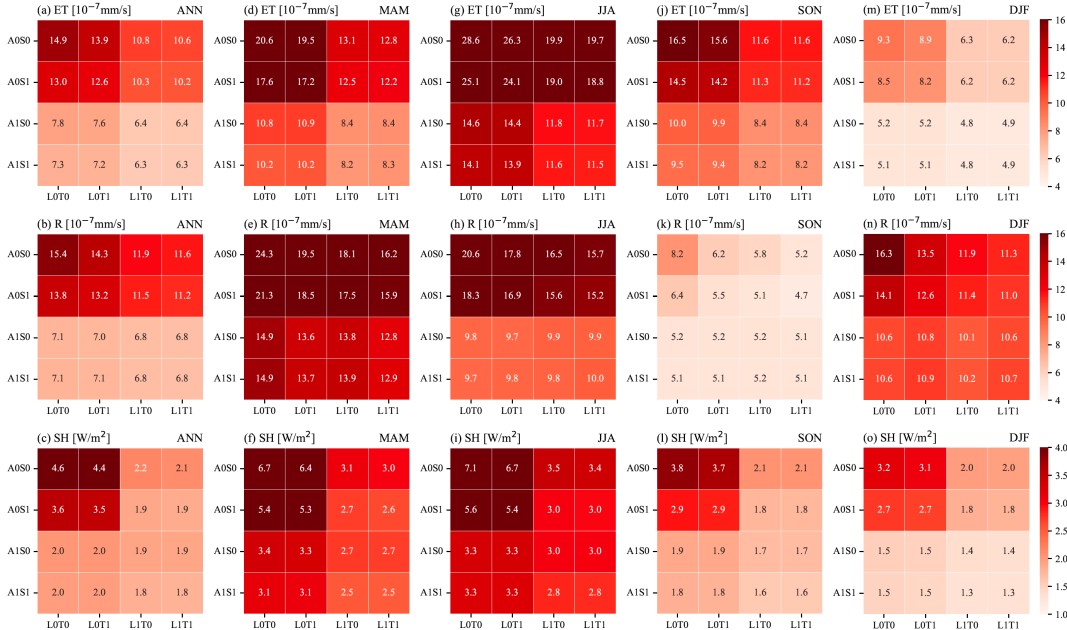

Figure 8. CONUS averaged absolute difference of SD between 16 ELM experiments and ERA5-
Land reanalysis for the annual (1st column) and seasonal (2nd – 5th column) climatological mean
of ET (top panel), R (middle panel), and SH (bottom panel).

**4. Discussions**





ATM and LULC are the two most essential heterogeneity sources contributing to the spatial
variability of water and energy partitioning. Our results are consistent with Simon et al. (2020),
who found that the heterogeneous meteorological forcing is the primary driver for the spatial
variability of latent heat and sensible heat in WRF simulations. The Sobol' sensitivity index
averaged over the same region (a 100 km × 100 km domain centered at 36.6° N, 97.5° W) as Simon
et al. (2020) also indicates that ATM is the dominant heterogeneity source. Therefore, better
representation of ATM heterogeneity in climate models is crucial for modeling the water and
energy partitioning, especially for the three major components of precipitation, air temperature,
and longwave radiation. One approach of capturing ATM heterogeneity has been developed by
Tesfa et al. (2020) for downscaling the grid mean precipitation to topography-based subgrids for
land surface modeling. Besides ATM, LULC is the second most crucial heterogeneity source.
Notably, anthropogenic land use and land cover change has been shown to have large impacts on
land–atmosphere interaction, land surface hydrology, and associated extreme events (Findell et al.,
2017; Li et al., 2018, 2015; Swann et al., 2010; Zeng et al., 2017; Yuan et al., 2021; Pielke et al.,
2007). Therefore, the heterogeneity of LULC should also be well considered in climate modeling.

ATM and LULC show complementary contributions to the spatial variability of water and energy
partitioning spatially over CONUS and temporally in different seasons. Sobol' sensitivity analysis
is a standardized quantification of the relative importance of different heterogeneity sources. The
sum of the Sobol' indexes for the four heterogeneity sources roughly equals one. As the two
dominant heterogeneity sources, ATM Sobol index and LULC Sobol' index dominate the sum of
all Sobol' indexes. Hence, they show complementary patterns spatially (Figure 5) and temporally
(Figure 6). In addition, ATM and LULC show complementary contributions across different



climate zones. The Budyko's aridity index (BAI, Budyko 1974), which is the ratio of annual net
radiation to the product of the latent heat of water vaporization and the annual precipitation, was
calculated using the outputs from EXP16. From humid (low BAI) to arid climate (high BAI), a
decreasing fraction of the CONUS region is dominated by ATM in determining the ET/P spatial
variability (Figure 9a). At the same time, LULC shows an increasing contribution to the ET/P
spatial variability with BAI. The spatial variability of energy partitioning exhibits even more
complementarity between the ATM and LULC contributions from arid regions to humid regions
(Figure 9c). In more arid regions limited by water, EF spatial variability is much more dominated
by heterogeneity of ATM, likely through the heterogeneous precipitation, but in humid regions
limited by energy, LULC dominates the EF spatial variability through its influence on surface
albedo and surface roughness.

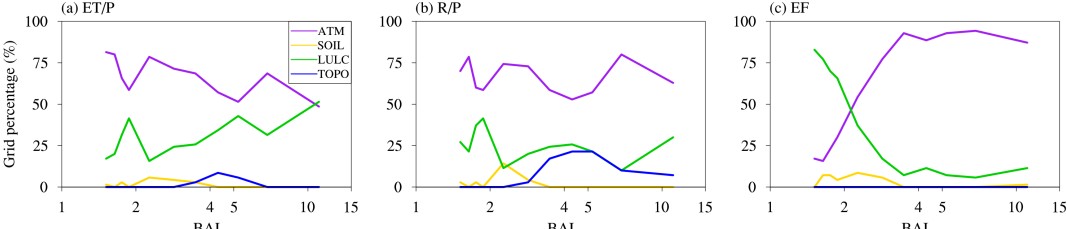


Figure 9. The grid percentage of dominant heterogeneity sources along with Budyko's aridity

index. A higher aridity index means more arid.


SOIL and TOPO show relatively small impacts on the spatial variability of water and energy
partitioning. However, TOPO has a dominant influence on the R/P spatial variability over the
transitional zone (Figure 9b) of central CONUS located between the arid western CONUS and the
humid eastern CONUS (Figure 4). SOIL shows some dominant effects on the spatial variability of
water and energy partitioning over a small proportion of humid regions (orange curve in Figure 9).





The heterogeneity in SOIL and TOPO was derived from coarse resolution data at 0.125°×0.125°
spatial resolution, which could be a possible reason for the minor SOIL and TOPO effects. Singh
et al. (2015) found that CLM4.0 did not show much improvement when model resolution increased
from ~100 km to ~25 km but improvement was noticeable at finer 1 km resolution. Additionally,
exclusion of lateral subsurface flow in ELMv1 could also lead to underestimation of the
contributions from SOIL and TOPO.

The current study excluded a few land surface processes that have been included in LSMs in the
last decade, limiting our ability to assess the role of land surface heterogeneity in spatiotemporal
variability of energy and water partitioning. For example, the hillslope processes of lateral ridge-
valley flow and the insolation contrasts between sunny and shady slopes are crucial for land surface
modeling (Fan et al., 2019; Taylor et al., 2012; Clark et al., 2015; Scheidegger et al., 2021), but
they are neglected in this study. Sean et al. (2019) incorporated the representative hillslope concept
into the CLM5, and they found that subgrid hillslope process induced large differences in
evapotranspiration between upland and lowland hillslope columns in arid and semiarid regions.
Krakauer et al. (2014) suggested that the magnitude of between-grid groundwater flow becomes
significant over larger regions at higher model resolution. Xie et al. (2020) also demonstrated the
importance of groundwater lateral flow in offsetting depression cones caused by intensive
groundwater pumping. Fang et al. (2017) compared the ACME Land Model (the earlier version of
ELM) and the three-dimensional ParFlow variably saturated flow model (Maxwell et al., 2015),
underscoring ELM limitation in capturing topography's influence on groundwater and runoff.
Additionally, topography also significantly influences insolation, including the shadow effects and
multi-scattering between adjacent terrain. Hao et al. (2021) implemented a sub-grid topographic



parameterization in ELM, which improves the simulated surface energy balance, snow cover, and
surface air temperature over the Tibetan Plateau. The inclusion of plant hydraulics has also shown
essential improvements in water and carbon simulations under drought conditions (Li et al., 2021;
Fang et al., 2021), which should also be considered in future research, especially as vegetation
may experience more hydroclimate drought stress in projected future climate conditions (Yuan et
al., 2019; Xu et al., 2019; Li et al., 2020). The subgrid downscaling of atmospheric forcing (Tesfa
et al., 2020), which could further enhance the representation of heterogeneity effects on water and
energy simulations, is also unaccounted for in this study.

**5. Conclusions**
This study comprehensively investigated the impacts of different heterogeneity sources (i.e., ATM,
LULC, SOIL, TOPO) on the spatial variability of water and energy partitioning over CONUS.
Two sets of experiments were conducted based on different combinations of spatially
heterogeneous and homogeneous datasets. Based on the first set of 16 experiments, Sobol' total
sensitivity analysis were utilized to identify the relative importance of the four heterogeneity
sources. The second set of 13 experiments were further used to assess the influence from individual
components of ATM, SOIL, and TOPO. Our results show that ATM and LULC are the two
dominant heterogeneity sources in determining the spatial variability of water and energy
partitioning. Their heterogeneity effects are spatially complementary across CONUS, and
temporally complementary across seasons. The complementary contributions of ATM and LULC
reflect the overall negligible impacts of SOIL and TOPO, but the complementarity also reflects
physically the clear demarcation of climatic zones across CONUS, featuring the arid, water-limited
western CONUS dominantly influenced by ATM (precipitation in particular) and the humid,



energy-limited eastern CONUS dominantly influenced by LULC. In the transitional climate zone
of central CONUS, TOPO shows some dominant influence on the R/P spatial variability. The
overall most essential components for ATM (precipitation, temperature, and longwave radiation),
SOIL (soil texture and soil color), and TOPO (Fmax) were also identified. Comparison with
ERA5-Land reanalysis reveals that accounting for more sources of heterogeneity improved the
simulated spatial variability of water and energy fluxes, although such improvements tend to
saturate as more heterogeneous sources were added.
The relative importance of different heterogeneity sources quantified in this study is useful for
prioritizing spatial heterogeneity to be included for improving land surface modeling. We note,
however, that the present assessment is limited by how well the input datasets capture the
spatiotemporal heterogeneity and how well the land surface model represent processes such as
hillslope hydrology and topographic effect on solar radiation that are influenced by land surface
heterogeneity. This motivates the use of more process-rich models such as distributed or three-
dimensional subsurface hydrology models to provide benchmarks of the relative importance of
heterogeneity sources to help prioritize future development of land surface models to improve
modeling of energy and water fluxes.



*Code and data availability*. The source code of ELMv1 is available from
https://github.com/E3SM-Project/E3SM (last access: September 2020); NLDAS-2 forcing is
available from https://ldas.gsfc.nasa.gov/nldas/v2/forcing; SOIL and TOPO related datasets are
downloaded   from   https://svn-ccsm-inputdata.cgd.ucar.edu/trunk/inputdata/lnd/clm2/rawdata/;
LULC related datasets are from Ke et al. (2012); ERA5-Land reanalysis is available from:
https://cds.climate.copernicus.eu/cdsapp#!/dataset/reanalysis-era5-land-monthly-
means?tab=overview.

*Author contributions*. LCL designed and conducted the experiments, analyzed model outputs, and
drafted the manuscript. GB designed the study, interpreted the results, and improved the
manuscript. LRL contributed to the interpretation and discussion of results and improvement of
the manuscript.

*Acknowledgments*. This research was conducted at Pacific Northwest National Laboratory,
operated for the U.S. Department of Energy by Battelle Memorial Institute under contract DE-
AC05-76RL01830. This study is supported by the US Department of Energy (DOE) Office of
Science Biological and Environmental Research as part of the Regional and Global Model
Analysis (RGMA) program area through the collaborative, multi-program Integrated Coastal
Modeling (ICoM) project. This study used DOE's Biological and Environmental Research Earth
System Modeling program's Compy computing cluster located at Pacific Northwest National
Laboratory.

*Competing interests*. The authors declare that they have no conflict of interest.



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
