# Peer review of "Spatial heterogeneity effects on land surface modeling of water and energy partitioning"

_Geoscientific Model Development, 2022_

## Author Response (AR1)

**Responses to comments <gmd-2022-4>**

Dear editors and reviewers:

We first want to thank you for your constructive comments, which are beneficial to improving our manuscript's quality. The revised manuscript includes the following major updates:

- Based on the comments of Reviewer 1 and to make the paper more concise, we have moved parts of the explanations of spatial patterns (section 3.2) and seasonal variations (section 3.3) of Sobol' total sensitivity index to Appendix B and Appendix C, respectively. We have also added the analysis of SD's difference between ELM simulations and ERA5_land reanalysis datasets in section 3.5.

- Based on the comments of Reviewer 2, we have added the analysis of Sobol' first-order index. Section 2.4 is updated, including a detailed description of different Sobol' indices, and a new diagram to better demonstrate the calculation of the Sobol' indices. Additional analysis is presented in sections 3.1 and 3.2. Lastly, we have also added the explanations for the seasonal variations of Sobol' total sensitivity indices for different heterogeneity sources in section 3.3.

The point-by-point responses to specific comments are provided below in blue font. All the line numbers listed below correspond to those of the revised manuscript (the clean version). We hope our modifications address the concerns raised in the last round of review and look forward to your decision on the publication of this manuscript.

Sincerely,

Lingcheng Li (on behalf of all authors)

*Editor*

(1) Paragraph 232-242 At the beginning of the paragraph, please clarify that this paragraph describes an example of one of four $SI_{xi}$.

We clarified the text in this paragraph, and based on reviewer 2's comments, we have moved this paragraph to Appendix A in line 567.

(2) Lines 251-252 "0.125×0.125 resolution", It is a little confusable to understand why 0.125×0.125 resolution? How about attaching "which is consistent with our sensitivity tests"?

We have now added an explanation for the choice of 0.125°×0.125° resolution in line 275. The sentence is updated as "First, the monthly data was regridded using the ESMF regridding tool via the first-order conservative interpolation to 0.125°×0.125° resolution, which is consistent with the resolution of our sensitivity experiments."

*Review 1*

This study investigated the effects of spatial heterogeneity in atmospheric forcing, land use and land cover, soil properties, and topography on the modeling of evapotranspiration, runoff, and their components (i.e., canopy evaporation, ground evaporation, transpiration, surface and subsurface runoff). The design of the numerical experiments is reasonable, the methods are innovative, and the results are insightful. With the above considerations, I suggest a publication with several clarifications.

Thank you for your encouraging comments.

Detailed comments

- L121--L124: Better put them in the Methodology section.

The methodology section 2.3 contains a detailed description of these experiments. Here we only briefly summarize these two sets of experiments in order to explain their corresponding roles. So, we would like to keep these two sentences.

- L135, 2013a Li et al., 2013: typos?

The typo has been fixed (line 137).

- L136--L140: What is the purpose of these statements? They duplicate the discussion. On the other hand, more descriptions of ELM on the used parameterizations are desirable.

We have removed these sentences in the revised manuscript. We have briefly described the use of surface data in the ELM parameterizations in section 2.2.1 (line 141–157)

- L166--L167: How did you resample soil properties? How soil texture and organic matter are upscaled, and how soil color is downscaled?

The ESMF regridding tool was used for dataset resampling. For the soil percentages of sand and clay, and soil organic matter (original resolution at 0.083°), we applied the ESMF regridding tool with the first-order conservative interpolation to do the upscaling. For the soil color, its original resolution is 0.5° and was downscaled using ESMF regridding tool with the nearest neighbor interpolation.

We have updated the sentences in line 163–167, " The SOIL, TOPO, and LULC were first mapped from their original resolutions to 0.125°×0.125° resolution, using the Earth System Modeling Framework (ESMF) regridding tool. Specifically, the first-order conservative interpolation was used for upscaling dataset (e.g., soil texture), while the nearest neighbor interpolation was used for downscaling dataset (e.g., soil color)."

- L199--L209: The calculation of Sobol's sensitivity index is still a bit confused to me. Assuming that X is the 30-year monthly value, SENSITIVITY(X) is the Sobol's sensitivity index of X, and 30-YEAR-SEASONAL-AVERAGE(X) is the seasonal average of X, did you calculate the index as SENSITIVITY (30-year-seasonal-average(X)) or 30-year-seasonal-average (SENSITIVITY(X)) ?

The calculated index corresponds to "SENSITIVITY (30-year-seasonal-average(X))".

In line 203, we made a slight modification. "First, the 30-year monthly, seasonal, and annual climatological means were calculated for each experiment at 0.125°×0.125° resolution for the five variables of interest (i.e., P, ET, R, LH, and SH)."

Then we described the calculation of the SD in line 207. "Third, the standard deviations (SDs) of **these variables' climatological mean** were calculated for each 1°×1° region from its embedded 64 0.125°×0.125° grids."

And finally, these SDs, which are calculated based on the variables' 30-year climatological mean, will be used for the sensitivity analysis. In line 208, "These 1°×1° resolution SDs of the first and second set of experiments were used in following analysis".

- Table 4: Did you first propose this approach?

The Sobol' total sensitivity index calculation follows Zheng et al. (2020). We clarify this in line 239 of the main text and line 571 of Appendix A.

In line 239, "For the calculation of $ST_{X_i}$: First, following Zheng et al. (2019), …"

In line 571, "(1) Calculation of $ST_{LULC}$ (Table A1): Following Zheng et al. (2019), …"

- L260: It would be to compare the spatial variability with the temporal variability or the mean value of ET/P.

It is the comparison of spatial variability, not the temporal variability, of the annual climatological mean of ET/P. As mentioned in line 203–209, (1) we calculated the 30-year annual climatological mean of ET and P at 0.125°. (2) then, we calculated ET/P at 0.125°. As they are climatological mean values, our study did not consider the temporal variability. Then (3) the standard deviations (SD) of these variables' climatological mean were calculated for each 1°×1° region from its embedded 64 0.125°×0.125° grids. Therefore, in this study, we only analyzed the spatial variability based on ET/P's climatological mean (including annual, seasonal, and monthly climatological mean) values instead of its temporal variability.

- L305--L328 and Figure 5: I did not get the objective of these contents. Since ATM and LULC are dominant, it is quite natural that their contributions to the spatial variability are complementary. Deletion seems fine and would make the paper concise.

As suggested, we have removed this paragraph from the main text. To be consistent, we have also removed the related seasonal explanation paragraph (initially the second paragraph in section 3.3) from the main text. However, we still believe these explanations are valuable for understanding the spatial patterns and seasonal variations of Sobol' total sensitivity index. Therefore, we have moved these two paragraphs to Appendix B (line 589) and Appendix C (line 617), respectively.

- L392-L395: Do the interplay between the spatial variabilities in ATM, LULC, SOIL, and TOPO increase or decrease the overall variability?

Based on the suggestion from reviewer 2, we also calculated Sobol' first order sensitivity index and the interaction effect index (i.e., the sum of higher order sensitivity indices). The heterogeneity effects from ATM and LULC are contributed mainly by the first order sensitivity index (line 305 and Figure 4). The overall interaction effects are positive, which means the interplay between different heterogeneity sources increases the overall variability compared to their first-order direct effect. But for a specific Sobol' higher-order index ($>2^{nd}$ order), it may be positive (increase overall variability) or negative (decrease overall variability).

- Section 3.5 and Figure 8: It is interesting to see the comparison between ERA5-Land and the 0.125-degree ELM simulations. Since the ERA5-Land atmospheric forcing is interpolated from the ~31-km ERA5 data with the consideration of elevation dependency (doi:10.5194/essd-13-4349-2021) rather than upscaled from the finer-scale observations, the spatial variability from ERA5-Land would be smaller than that from ELM. In Figure 8, it would be better to show the difference rather than absolute difference to check this.

We have added Figure S9 for the SD's difference between the four ELM experiments and ERA5-Land reanalysis (i.e., ELM – ERA5_Land). We have updated line 443–446, "Overall, the ELM simulated ET and SH have smaller SDs than those of ERA5_Land (Figures S9d and S9l). Meanwhile the simulated R has larger SD especially in the western US mainly due to ATM (Figures S9e vs. S9g)."

*Reviewer 2*

This manuscript examines the relative importance of four heterogeneity sources on the spatially aggregated variability of some modeled components of the water and energy partitioning over CONUS. The study is appropriately motivated and, overall, well written. The presented methods are adequate and innovative, and the results reasonable. I suggest publication after some minor revisions and clarifications.

Thanks for your inspiring comments.

L136 – L140: Since this study did not use the described model developments, I recommend removing this paragraph. Instead, details on the methods used to characterize the land surface heterogeneity within ELM for this study would be appreciated.

As suggested, we have removed the description of model developments not used in this study.

In the introduction, we mentioned that ELM uses a "tile approach" to account for spatial heterogeneity (line 53-54), and thus do not want to repeat the same content here in the methodology section 2. Additionally, in the discussion section 4, we described the ongoing heterogeneity related development in ELM (line 210).

L166 – L167: Please clarify the methods used to resample the original datasets to 0.125°.

The ESMF regridding tool was used for dataset resampling. For the soil percentages of sand and clay, and soil organic matter (original resolution at 0.083°), we applied the ESMF regridding tool with the first-order conservative interpolation to do the upscaling. For the soil color, its original resolution is 0.5° and was downscaled using the ESMF regridding tool with the nearest neighbor interpolation.

We have updated the sentences in line 163–167, " The SOIL, TOPO, and LULC were first mapped from their original resolutions to 0.125°×0.125° resolution, using the Earth System Modeling Framework (ESMF) regridding tool. Specifically, the first-order conservative interpolation was used for upscaling dataset (e.g., soil texture), while the nearest neighbor interpolation was used for downscaling dataset (e.g., soil color)."

L251 – L252: Please clarify the methods used to resample.

The ESMF regridding tool was used for dataset resampling.

Line 274–276 is updated as "First, the monthly data was regridded using the ESMF regridding tool via the first-order conservative interpolation to 0.125°×0.125° resolution, which is consistent with the resolution of our sensitivity experiments"

Section 2.4: Have you also considered analyzing the first-order Sobol sensitivity index? The first-order index would measure just the direct effect of each heterogeneity source on the variance of the model. Compared to the total sensitivity, the first order index would explain the importance of the interactions between heterogeneity sources (e.g., topography and atmospheric forcing; soils and topography).

Thanks for your insightful suggestion. In the revised manuscript, we have calculated Sobol' first-order sensitivity index and the interaction effect index (i.e., the sum of higher-order indices). Therefore, we updated the following parts:

(1) In section 2.4 (line 220), we have added descriptions for Sobol' indices calculation, and a schematic Figure 2 to better explain the computation of these Sobol' indices.

(2) We have also moved the demonstration table of Sobol' indices calculation (Table 4 in the original submission) to Appendix A as Table A1, and added calculation demonstration Table A2 and Table A3 for the first-order sensitivity index and interaction effect index in Appendix A (line 567).

(3) We have updated Figure 4, by adding the first-order index (Figure 4b) and its contribution fraction to the total sensitivity index (Figure 4c) and corresponding spatial patterns in Figure S4 and Figure S5. We have also added Figure S3 about the interaction effect index. Therefore, the analysis has also been updated (line 304–321, line 342–347).

(4) Generally, high values of total sensitivity indices are mostly contributed by its first order sensitivity index. Therefore, our analysis is still based chiefly on Sobol' total sensitivity index, and the analysis and conclusion are similar to the original submission. The related

content in the abstract, discussion (line 458 and line 500), and conclusion (line 540 and line 544) were accordingly updated.

L361 – L370: A more comprehensive analysis of the factors explaining the seasonal variation of the importance of heterogeneity sources would be appreciated here. For instance, what drives the increased relevance of topography in zones of the East in Summer and Fall.

We selected a grid cell to demonstrate the changes of the dominant heterogeneity source in eastern US in summer and fall. We added Figure S7. ATM induced R's SD shows large seasonal variations, with large values in spring and winter but small values in summer and fall. Therefore, ATM is the dominant heterogeneity source in spring and winter. Even though TOPO and SOIL induced R's SD show small seasonal variations, they have large values in summer and fall, respectively. Therefore, TOPO and SOIL dominate R's spatial variability in summer and fall, respectively.

The updated text is added in line 383–390.

Table 5: I recommend moving this table to Supplementary material and including Figure S5 in the main text (primarily, since L361 to L370 mainly focus on analyzing maps).

As suggested, we have moved the original Figure S5 to the main text as Figure 8 and updated the corresponding figure number in section 3.4.

The seasonal variation analysis in line 373–383 is mainly based on table 5. So, we would like to keep this table here.